# Bridging the Connection between Fluency in Reading and Arithmetic

**DOI:** 10.3390/bs14090835

**Published:** 2024-09-18

**Authors:** Reut Balhinez, Shelley Shaul

**Affiliations:** Edmond J. Safra Brain Research Center for the Study of Learning Disabilities, Department of Learning Disabilities, Faculty of Education, University of Haifa, Haifa 3498838, Israel; reut136@gmail.com

**Keywords:** reading fluency, arithmetic fluency, executive functions

## Abstract

This study examines the contribution of early executive functions (EFs) in the association between fluency in reading and arithmetic. Kindergarten children (N = 1185) were assessed on executive functions skills and on reading and arithmetic fluency in Grade 1 and Grade 3. The analysis revealed that beyond the connection within each domain there is a unidirectional effect between fluency measures, with Grade 1 reading fluency significantly influencing the development of arithmetic fluency in Grade 3. Furthermore, the findings indicate that kindergarten EFs significantly contribute to arithmetic fluency at both time points and to reading fluency in the first grade. Early EF skills also emerged as significant contributors to the associations between fluency performance in reading and arithmetic, suggesting that the influence of EFs extends beyond individual academic domains. These findings have implications for understanding the cognitive mechanisms that underlie the relations between these academic skills.

## 1. Introduction

### 1.1. Fluency in Reading and Arithmetic: Development and Cross-Domain Association

Fluency in reading and simple arithmetic calculations are fundamental cornerstones required for overall academic development, daily life activities, and success [1,2]. The measure of fluency (i.e., a combination of accuracy and speed) often serves as an indicator for the level of efficiency and automaticity in word reading [3,4,5,6] and simple single-digit arithmetic facts calculation, such as 4 + 5 = 9 and 6 × 3 = 18 [7,8,9].

Fluency acquisition, across both academic domains, relies on one-by-one serial coding. At this initial level, reading depends on the decoding of written graphemes (letters) into phonemes (sounds) [10]. With time, practice, and repetitive print exposure, the reader becomes capable of automatically processing and retrieving larger orthographic units, such as whole words [11]. In a similar manner, calculation is based on serially reciting number words using concrete representations (such as fingers and objects) and verbal counting [12,13]. Gradually, efficient counting and decomposition strategies strengthen the association between the arithmetic equation and its correct answer [13], which, in turn, increases fluent and automatic memory-based retrieval of facts [14]. Well-established fluency promotes learning by freeing up cognitive resources for the execution of complex tasks, such as reading comprehension, e.g., [15] and algorithmic and multi-digit computation [16]. Parallel to the developmental course of typical fluency acquisition, there is also strong evidence for high rates of learning disabilities, which co-occur in both academic domains; these children with disabilities may not achieve fluency in either or both domains, e.g., [17,18].

In recent years, a growing body of research on the development of fluency in reading and arithmetic has revealed substantial cross-domain correlation [19,20], which is also manifested in a high degree of overlap between these skills [21,22,23]. For instance, a cross-sectional research study by Balhinez and Shaul (2019) [19] revealed a significant and consistent moderate correlation between fluency in reading and arithmetic among first-, second-, and third-grade Hebrew-speaking students. Similar results were found in a series of longitudinal studies in elementary education across different languages and cultural backgrounds, e.g., Chinese: [24], English: [25], Greek: [26], Finnish: [20,27,28], through the secondary school years, Germany: [29].

However, the characteristics of orthographies may also influence the dynamics observed in the cross-domain relationship between reading and arithmetic skills, specifically the development of reading proficiency, which varies across languages [30]. In shallow orthographies characterized by consistent grapheme-phoneme correspondences, such as Hebrew, reading proficiency tends to develop more readily and rapidly [31] compared to deep and inconsistent orthographies such as English [32]. The developmental paths of reading fluency, influenced by orthographic complexity, highlight the need for a language-specific approach in studying cross-domain relations between reading and arithmetic skills. Combined with the overview of the correlational studies, there is clear evidence for a bidirectional effect between early reading and mathematics, particularly during the first years of elementary school [33,34,35,36,37]. However, the picture is far from conclusive, as findings on the extent of these mutual relations are mixed. Some have reported higher associations for early reading performance to later mathematic outcomes [33,36,37], while others have found stronger predictive power of early mathematics for later reading performance [35,38,39].

Potential explanations for the observed associations between both forms of fluency typically refer to a common genetic factor of timed measures [40,41], joint neural-brain regions activated during both math and reading task performance [42], and shared common underling cognitive mechanisms such as executive functions (EFs) [19,22,23,43].

Despite the fact that large bodies of literature have confirmed the association between reading and math skills among young students, several fundamental questions still remain unclear. First, as most of the reviewed studies investigated a wide range of reading and arithmetic-related skills, little is known about the cross-domain developmental relation of fluency outcomes. Even far less systematic research is available concerning the crossover effects between both types of fluency across different stages of acquisition. Second, given the joint reliance of reading and arithmetic on early cognitive mechanisms, such as EFs, an important unaddressed question is whether and to what extent earlier executive functions contribute to (i.e., account for) the correlation between reading and arithmetic fluency at later ages.

### 1.2. Executive Functions and Development of Fluency

Higher-order cognitive abilities are intensively involved in knowledge acquisition and the learning of new skills. These domain-general processes, which typically are referred to as executive functions, are responsible for self-regulation, goal-oriented behavior [44], and complex problem-solving [45,46]. A framework suggested by Miyake and his colleagues (2000) [47] refers to three main interrelated subcomponent skills of EFs, namely the ability to suspend an automatic response (inhibitory control), the ability to temporarily store and manipulate information (updating), and the ability to shift between different tasks and operations (shifting). An additional framework was suggested by Diamond (2013) [48], which includes inhibition, working memory, and flexibility [48]. While there are some differences between Miyake’s and Diamond’s models, this study is based on Diamond’s model, which includes broader concepts. Previous latent variable studies have demonstrated that executive-related abilities mostly tap into a single one-factor construct (unity) among early childhood samples [49] and cluster into separate constructs (diversity) over the course of development [50,51].

The childhood years represent a critical period in the growth of cognitive skills and therefore are perceived to have long-term effects on future school adjustment and scholastic achievement [52]. The extent to which EFs operate and contribute to reading and arithmetic appears to vary across the early childhood years [53]. Thus, less proficient learners rely more on cognitive resources compared to more skilled students [54]. In the early phases of reading acquisition, EFs might support the suppression of incorrectly deciphered words [55], the simultaneous storage and integration of phonological information into a single word [56], and efficient switching between different meta-linguistic aspects of spoken and printed words [57]. Similarly, EFs are thought to facilitate arithmetic calculations by deactivating inappropriate procedures and/or incorrect solutions [58], supporting flexible shifting between problems and procedures, and facilitating the simultaneous execution of strategies [59,60].

There is a considerable amount of empirical evidence regarding the contribution of early EF skills to later reading [61,62] and arithmetic outcomes [45,58,59], especially during the first years of schooling [63,64,65]. However, to date, almost all research has focused on the predictive effect of early EFs on overall academic performance; a specific domain (literacy/mathematics) such as fluency, which is considered a core skill, has not been properly investigated. Given the strong relevance of EFs to academic performance and the tremendous growth of these components in young children, it is highly important to understand the nature of the long-term changes in the predictive role of EFs on fluency in reading and arithmetic across the major developmental periods of their acquisition.

As EFs seem to play a significant role in the development of both mathematics and reading, researchers are increasingly interested in the potential direct and indirect effects of EFs on the associations between learning-related skills.

### 1.3. The Contribution of EFs to the Connection bewteen Reading and Mathematical Abilities

Alongside the fairly well-established contribution of EF skills to academic outcomes, some longitudinal datasets also support the reverse direction of influence from academic performance to EF skills. Specifically, a recent line of studies has shown that early EFs were positively predicted by various literacy skills [66,67,68] and math skills [66,69]. The reciprocal co-developing associations between academic achievements and EF skills are also manifested in the mutual growth effects across these two domains. Thus, children with better EF skills tend to show a stronger relative increase in the development of early academic skills and vice versa [34,70,71,72,73]. The theoretical framework for understanding the developmental effects of academic performance on EFs posits that academic-related activities create opportunities to practice multiple cognitive processes [74,75]. Thus, reading acquisition exercises higher-order skills, such as segmentation and integration, which involve the serial processing of alphabetic symbols into a whole word [34]. Similarly, early computational activities promote the ability to simultaneously hold and operate relevant information while applying the appropriate strategies to solve arithmetic problems [38,76].

Given the co-mutual beneficial influences between EFs and academic development, and the bidirectional relationship between reading and arithmetic, there is an inevitable question regarding the potential role of EFs as a contributor to the relationship between both types of fluency. To date, however, the few studies which have examined EF skills as a mediator in academic-related skills focused on the association between early and later arithmetic performance [77,78]. Both longitudinal studies revealed a consistent mediating effect of early EF skills in the developmental associations between mathematics performance from kindergarten through the first year of primary school among children with typical achievements and at risk for mathematics disability.

To the best of our knowledge, only one recent study, by Braak et al. (2022) [38], has investigated the role of EFs as a mediator in the cross-domain association between mathematics and reading. Consistent with the reported results in research reviewed previously, first-grade EFs significantly explain the developmental correlations between kindergarten mathematics and fifth-grade mathematics and reading achievements. However, the study by Braak et al. (2022) [38] is limited by its methodological weaknesses, such as the use of only one test to measure multiple EF subskills and the inclusion of a variety of academic skills, which range in complexity and cognitive requirements. Thus, the main goal of our study is to elucidate and directly investigate whether and how early EF skills, before school transition, contribute to the relationship between fluency in reading and arithmetic in the first and third grade. Understanding the nuanced pattern of the cross-domain associations between those key fundamental learning skills using a longitudinal systematic analysis might provide unique insights into future research on assessment and educational intervention.

### 1.4. The Present Study

The purpose of this study is to examine the developmental nature of the relationship between fluency in reading and arithmetic and the role of early EFs in these cross-domain associations. Three research questions guided the work:(a)How does the cross-domain developmental relation between fluency in reading and arithmetic evolve across the major stages of their acquisition, and what is the direction of influence between these two skills?

Considering the robust relations between reading and arithmetic [19], it was hypothesized that fluency skills in both academic domains would show a consistent, significant correlation. However, considering the mixed findings in the literature regarding the strength and direction of the mutual cross-domain effect between reading and arithmetic [33,38] and the potential impact of orthographic transparency on reading fluency acquisition [30], we assume that there may be variability in the relative contribution between both types of fluency across the different developmental stages.

(b)What is the relative contribution of kindergarten EFs in the development of fluency in reading and arithmetic? And to what extent do earlier executive functions contribute to (i.e., account for) the connection between reading and arithmetic fluency at later ages?

We anticipate that the predictive power of EFs on fluency outcomes will vary over time, reflecting distinct developmental trajectories in reading and arithmetic skills. This hypothesis aligns with the notion that early EFs play a significant role in the development of new skills learned [79], such as fluency, in both reading and arithmetic during the early stages of acquisition. Given the high transparency of Hebrew orthography, which facilitates early reading fluency [31], we expect a relatively weaker impact of early EFs on reading fluency compared to arithmetic fluency during the advanced stages of formal schooling. In addition, due to the significant contribution of EF to both academic domains, we assume that they will contribute to the connection between them as well.

## 2. Method

### 2.1. Participants

A total sample of 1142 Hebrew-speaking children was recruited from 128 kindergartens in and around the greater Haifa region of northern Israel and later followed into their elementary schools. The sample represented diverse socioeconomic backgrounds and included both regular and religious kindergartens, excluding ultra-orthodox and special education institutions due to curriculum differences. The total sample of 1142 children participated in all tests during kindergarten (time 1), 798 participated in first grade (time 2), and 718 participated in third grade (time 3) of elementary school. The average age of the children at the start of the study was 71.14 months (SD = 5.87 months). All participants were typically developing native Hebrew speakers with normal IQ and no neurological disorders. All statistical analysis took the missing data into consideration.

### 2.2. Measures

#### 2.2.1. Executive Functions in Kindergarten

##### Inhibition

***Head-Toes-Knees-Shoulders (HTKS)*** [80] was used as a behavior regulation measure. The task, which is designed for young children, is mainly tapping into the ability to integrate several cognitive processes, such as attention, working memory, and inhibitory control [81]. The children were instructed to do the opposite of the examiner’s instructions (e.g., touch their heads when told to touch their toes and vice versa). The first part consisted of two rules (head or toes), and the second part added two novel commands (knees or shoulders). After practice trials followed by the examiner’s feedback, the tests, which included 20 trials (10 trials for each part), were performed. Two points were given for each correct response at the first attempt. Self-correction from the incorrect to the correct response was given one point, and an incorrect response was given 0 points. The final total score for the 20 test items ranged between 0 and 40. The test-retest reliability for this test is 0.75.

##### Working Memory (WM)

***Visual-Spatial WM*** the Corsi frog backward test, adapted from the DEST-2 [82], was used. The task uses a colored printed card with seven lily pads, arranged in random order, along with a small plastic toy frog. The children are required to watch the frog’s jumping on the lily pads and then copy the frog’s jumps in reverse order. A practice trial was administered, followed by instructive feedback. The difficulty of the tasks gradually increases, starting with blocks of a two-jump sequence and gradually increasing to blocks of seven jumps; each block contained two sequences. The test is concluded when the children fail to correctly reproduce the sequence backwards on two consecutive trials in the same block. The capacity score is calculated based on the longest list length correctly recalled. Cronbach’s alpha indicating reliability is 0.81.

***The digit span backward test*** [83] consists of a sequence of random numbers presented verbally, beginning with a sequence of two digits and increasing in length to blocks of seven digits; each block contained two sequences. The children are instructed to repeat the digits in reverse order, from the last number they had heard to the first one. The test is discontinued when two mistakes occur with the same block. The final score is calculated based on the longest list correctly recalled. The test-retest reliability for this test is 0.72.

***The children’s size-ordering test (CSOT)*** [84] contains a list of common objects (e.g., window, book) read aloud at a rate of one item per second. Children are required to reproduce these verbally in order of size, from smallest to largest (e.g., book, window). The test comprises six blocks with two items per block. The level of difficulty increases gradually, beginning with two items to a maximum of seven items. The test is discontinued when two mistakes occur in the same block. Granvald and Marciszko (2016) [85] reported split-half reliability for the CSOT of 0.63.

##### Cognitive Flexibility

***The sorting cards-shifting task***, adapted from the Object Classification Task for Children [86], was used. As opposed to the original version of the OCTC, which uses plastic toys, in the present study cards were presented. A practice trial was administered to ensure that the children were capable of sorting using overall visual features. The test requires sorting of six printed pictures into different groups according to three possible dimensions: size (big or small), color (red or yellow), and shape (car or plane). The task consisted of three conditions differentiated by the level of assistance provided by the examiner. It starts with the first condition, and only if the child failed, the next condition is administered.

(1)Free generation condition, where the child is required to sort the pictures without any clues or assistance and verbally name the common features of the grouped pictures. Then the examiner mixes up the cards and instructs the child to sort them by different dimensions. This procedure was repeated until the child correctly sorted the cards according to all three predetermined groupings (i.e., size, color, and shape). The child got 4 points per correct sorting in this level.(2)Identification condition, in which the examiner groups the cards by the specific dimension(s) in which the child failed to sort them in the free generation condition and asks him or her to identify the sorting category. The child got 3 points per correct sorting in this level.(3)Explicit cueing condition, in which the examiner provides instructions on how to group the pictures by a specific dimension in which the child failed to classify them in the identification condition. The child got 2 points per correct sorting in this level.

The score was calculated according to the condition that the children performed the task, with a maximum of 12 points for the highest level. Cronbach’s alpha indicated a reliability for this test of 0.75.

##### Fluency Measures for the First and Third Grades

***Reading Fluency***, Reading fluency was measured using the Test of Word Reading Efficiency, adapted from the TOWRE test [87,88], as follows: The kamatz-patach subtest (a test which included words only with the vowel/a/-single-vowel test) was administered in the first grade, and the fully vowelized words subtest (a test which included words with all the vowels) was administered in the first and third grades. The subtests of oral reading accuracy and fluency contain a list of words increasing in difficulty and varying in frequency, length, and phonological and morphological structure. The children were requested to read the words aloud as quickly and accurately as possible. The total number of correct words read per 60 s was calculated.

***Arithmetic Fact Fluency*** First-grade arithmetic fluency performance was measured using addition and subtraction subtests (developed by the Safra team). Third-grade arithmetic fluency performance was measured using the WJ-III Math Fluency test [89]. In both tests, children are instructed to complete as many simple single-digit facts, skipping those they do not know. The total number of problems solved correctly is calculated. The reported reliability for the WJ-III Math Fluency test is between 0.90 and 0.93.

### 2.3. Procedure

Prior to the collection of the data, the required approvals were obtained from the Ministry of Education as well as the Ethics Committee of the university. In addition, consent forms were signed by the parents of the children examined. All the tests were administered to the participants individually during kindergarten time, but in a separate room, in two or three separate sessions of about 20 min each. In the first and third grades, the children were also tested individually in a separate room in two sessions of 45 min each. All tests were administered in random order. In order to obtain test-retest reliability for the HTKS and the backward digit span, these tests were administered again two weeks after the first time.

### 2.4. Data Analysis Plan

The analysis proceeded in four stages. First, confirmatory factor analysis was conducted to examine the construct validity of the kindergarten EF measures and the first-grade fluency tests in reading (kamatz and all-vowel word reading subtests) and arithmetic. The confirmatory factor analysis (CFA) was employed using AMOS software (version 27.0) to confirm the kindergarten EF and first-grade fluency constructs.

Second, descriptive statistics and bivariate correlations of the study variables and composite scores were calculated using SPSS 27.0. Third, structural equation modeling (SEM) was conducted using AMOS software 26.0 to test the predictive relation of early EFs to later fluency performance in reading and arithmetic across first and third grades. Finally, two-step hierarchal regressions were performed for each age group separately using SPSS 27.0. The first block examined the shared variance between reading and arithmetic fluency in each age group, and the second block examined how much of the residual variance is explained by the early kindergarten EFs.

## 3. Results

### 3.1. Confirmatory Factor Analyses of the Observed Executive Functions and Fluency in Reading and Arithmetic Variables

Data reduction was obtained by conducting a confirmatory factor analysis on the kindergarten EF items (HTKS, OCTC, CSOT, digit span backwards, and Corsi frog backwards), which included 981 children in kindergarten. The confirmatory factor analysis (CFA) confirmed the validity of the model (see Figure 1). This model provided a good fit for the data (χ2 = 6.852, df = 5, *p* = 0.232, NFI = 0.984, CFI = 0.995, TLI = 0.986, RMSEA = 0.018).

Based on this analysis, a new composite variable was built for the kindergarten EFs, which represent the average Z score of all the EF measures.

In the second CFA, we tested how well the observed variables reflected the fluency constructs in reading (kamatz and all-vowel word reading subtests) and arithmetic (addition and subtraction problem-solving subtests) in the first grade among 782 children (see Figure 2). The model provided a good fit for the data (χ2 = 0.213, df = 1, *p* = 0.644, NFI = 1.00, CFI = 1.00, TLI = 1.00, RMSEA = 0.000).

### 3.2. Correlations

Descriptive statistics (means, standard deviations, and maximum and minimum scores) at all-time points and correlations between measures are presented in Table 1.

Bivariate correlations were conducted to explore the possible relationships between the composite overall score of EFs and first- and third-grade fluency in reading and arithmetic (see Table 1). Significant correlations between EF composite scores and fluency tests throughout the examined grades ranged from 0.22 to 0.38.

The overall pattern of correlational extent between fluency measures varied from low to moderate. Specifically, fluency performance associations within the same academic domain were consistently moderate across grades. When comparing the cross-domain relationship between both types of fluency, correlations were moderate within grades (RF first grade and AF first grade r = 0.52, RF third grade, and AF third grade r = 0.51) and across grades (RF first grade and AF third grade r = 0.38, RF third grade, and AF first grade r = 0.45).

### 3.3. Predictive Relation of Early EFs to Later Fluency Skills in Reading and Arithmetic Structural Equation Modeling (SEM)

Structural equation modeling was used to examine the role of kindergarten EF skills in later reading and arithmetic fluency performance among first and third graders (see Figure 3). We began the analysis with the structural part of the model (the relationship between the latent variables and the fluency outcomes in 3rd grade) saturated. The model demonstrated good fit indices (χ2 = 86.412, df = 36, *p* < 0.001, NFI = 0.974, CFI = 0.984, TLI = 0.971, RMSEA = 0.034). An in-depth examination of the relationships between the study variables reveal nonsignificant predictive associations between kindergarten EF skills and reading fluency in the third grade as well as between arithmetic fluency in the first grade and reading fluency performance in the third grade. In order to reach a parsimonious model, these paths were removed.

A comparison of the models’ fit indicated that the chi-square was not significantly impaired as a result of path removal (Δχ2 = 3.17, df = 2, *p* = 0.205). The final pruned model, presented in Figure 3, shows a good fit for the data (χ2 = 89.570, df = 38, *p* < 0.001, NFI = 0.973, CFI = 0.984, TLI = 0.972, RMSEA = 0.034). The EFs predicted both types of fluency outcomes in the first grade, as well as later arithmetic fluency in the third grade. First-grade reading fluency is positively and significantly related to third-grade fluency performance in both academic domains. As anticipated, first-grade arithmetic fluency is additionally predictive of later arithmetic fluency skill in the third grade.

### 3.4. The Contribution of Early EFs to the Connection between Reading and Arithmetic Fluency

In order to examine to what extent early kindergarten EFs contribute to the shared variance between reading and arithmetic fluency, hierarchal two-step regressions were performed for each age group separately. The first block examined the shared variance between the reading and arithmetic fluency in each age group, and the second block examined how much of the residual variance is explained by the early kindergarten EFs.

#### 3.4.1. First Grade

Step 1: Reading Fluency and Arithmetic Fluency: A simple linear regression was performed to predict reading fluency based on arithmetic fluency. The model was statistically significant: F(1, 565) = 205.77, *p* < 0.001, R^2^ = 0.27. Arithmetic ability significantly predicted reading ability, β = 0.52, t(565) = 14.35, *p* < 0.001.

Step 2: Residuals and Executive Function: In the second step, a regression analysis was conducted to predict the residuals from the first model (representing variance in reading fluency not explained by arithmetic fluency) based on executive function. This model was also statistically significant: F(1, 565) = 19.31, *p* < 0.001, R^2^ = 0.03. Executive function significantly predicted the residuals, β = 0.18, t(565) = 4.39, *p* < 0.001.

In examining the opposite direction model (representing variance in arithemtic fluency not explained by reading fluency) based on executive function. This model was also statistically significant: F(1, 565) = 21.54, *p* < 0.001, R^2^ = 0.04. Executive function significantly predicted the residuals: β = 0.19, t(565) = 4.64, *p* < 0.001.

The results indicate that 27% of the variance in reading fluency is shared with arithmetic fluency. However, out of the total shared variance between the two fluencies, EF in kindergarten explains 8%. The unique contribution of EF to either reading fluency or arithmetic fluency in the first grade is clearly smaller (namely, 3.1% and 3.7%).

#### 3.4.2. Third Grade

Step 1: Reading Fluency and Arithmetic Fluency: A simple linear regression was performed to predict reading fluency based on arithmetic fluency. The model was statistically significant, F(1, 409) = 161.49, *p* < 0.001, R^2^ = 0.28. Arithmetic ability significantly predicted reading ability: β = 0.53, t(409) = 12.71, *p* < 0.001.

Step 2: Residuals and Executive Function: In the second step, a regression analysis was conducted to predict the residuals from the first model (representing variance in reading fluency not explained by arithmetic fluency) based on executive function. This model was also statistically significant: F(1, 409) = 3.92, *p* < 0.05, R^2^ = 0.01. Executive function significantly predicted the residuals, β = 0.10, t(409) = 1.98, *p* < 0.05.

In examining the opposite direction model (representing variance in arithmetic fluency not explained by reading fluency) based on executive function. This model was also statistically significant: F(1, 409) = 7.16, *p* < 0.01, R^2^ = 0.02. Executive function significantly predicted the residuals, β = 0.13, t(565) = 2.67, *p* < 0.01.

The results indicate that 28% of the variance in reading fluency is shared with arithmetic fluency. However, out of the total shared variance between the two fluencies, EF in kindergarten explains 3.5%. The unique contribution of EF to either reading fluency or arithmetic fluency in the third grade is smaller (namely, 1.0% and 1.7%).

The summary of these results is presented in Figure 4.

## 4. Discussion

The main aim of the present study is to investigate whether and how early EF skills affect the cross-domain relations between fluency in reading and arithmetic across different phases of skill acquisition. Novel aspects of the study include: (1) a systematic assessment of the longitudinal correlations and predictive relations of fluency in reading and arithmetic across the early elementary grades; (2) a unique examination of the relative contribution of kindergarten EFs to the development of fluency in both academic domains; and (3) an investigation of the role of early EFs in the connection between both fluency measures in each age group. To the best of our knowledge, this is the first study that has attempted to explain the cross-domain relationship between fluency in reading and arithmetic by examining the potential effect of common core EF components. In order to adequately capture the developmental nature of the complex relationships among these skills, a large sample of kindergarten children was followed from kindergarten to the third grade.

### 4.1. Cross-Domain Relations between Fluency in Reading and Arithmetic

Overall, the results of this study indicate that, consistent with past research, fluency performance in reading and arithmetic are significantly and positively correlated (ranging from weak to moderate). Despite the interrelations between these skills and previous evidence showing mutual effects between reading and arithmetic-related skills, e.g., [34,38,43], bidirectional predictive relations were not found in the current study. Accordingly, our analysis reveals that the cross-domain predictive role of fluency is unidirectional, as first-grade reading fluency predicted arithmetic fluency performance in the third grade. A longitudinal study with academically at-risk children [90] similarly found that only reading performance contributed to math growth from first through fourth grades. It should be noted that the reviewed studies are not directly comparable due to major differences in the analysis approach, measurement aspects (i.e., composite score on broad measures), and unique characteristics of the samples studied (typical achievers vs. at-risk for learning difficulties).

Another possible explanation for the observed role of reading fluency in the development of arithmetic fluency is based on the premise that mathematics, and arithmetic fluency in particular, is a language-based knowledge [91,92]. Thus, learning simple calculation comprises reading-related skills, such as phonological awareness required for the retrieval of numeric words, processing, and encoding of symbols in the long-term memory [93], verbal retrieval abilities such as rapid automated naming [94], and specialized vocabulary [26,95].

We also assume that the predictive interaction between fluency in reading and arithmetic might be further influenced by pedagogical aspects combined with specific features of the orthography examined. In the orthographically regular Hebrew language (i.e., grapheme-phoneme correspondence is both consistent and relatively easy to master), reading fluency is largely instructed and established by the end of the first grade [31]. By contrast, the formal instruction and practice of simple arithmetic calculations typically continue until the end of the third grade [96,97]. As Hebrew reading fluency increases faster and earlier compared to the ability to retrieve arithmetic calculations, it might be that fluent word reading performance relies on different skills and processes (such as metacognitive and/or metalinguistic components) than those required for the execution of simple calculations in the first year of primary school.

### 4.2. The Role of Early EF Skills in the Development of Fluency

In testing the long-term predictive role of kindergarten EFs in later fluency performance, we found a significant contribution of kindergarten EF components (which emerged as a single unitary construct) to first-grade reading and arithmetic fluency acquisition. On the other hand, by the third grade, early EFs are significantly connected only to arithmetic fluency performance but not reading fluency performance.

From a developmental perspective, the time specificity of the predictive effect of kindergarten EFs shared by fluency in both academic domains, which was obtained only in the first grade, might be associated with differences in the developmental timeline for the acquisition of fluency across both academic domains. Hence, the common significant importance of early EFs in fluency performance during the first year of formal school might indicate that both are at the initial stages of new skill learning. As noted in the introduction, in the beginning stages of fluency acquisition, the young learner performs complex activities, such as serial decoding of graphemes into phonemes while deciphering a new word and implementing calculation procedures (such as finger counting), which, in turn, lead to an increase in cognitive load. While reading fluency leads to the automatization of skillful actions and frees up cognitive resources by the third year of school, simple calculations still rely on EF processes, indicating that the retrieval of arithmetic facts has not yet reached an automatic level. These results strengthen the notion that the developmental trajectories of reading vs. arithmetic fluency are different and that the automatization process is domain-specific and not general.

### 4.3. Executive Functions as a Contributer to the Relationship between Fluency in Reading and Arithmetic

These findings, together with the other findings, suggest that while there is a substantial overlap between reading and arithmetic abilities in both the first and third grades, executive function plays a significant but small role in explaining part of this shared variance. The results emphasize that early EFs contribute to the connection between reading and arithmetic fluency in the first grade more than in the third grade; this may be due to the development of the EFs during the years as well as the engagement of more specific abilities during the years with the development of these skills as opposed to the early stages of the establishment of these skills.

It is important to note that while EFs in kindergarten contribute about 8% to the shared variance between both types of fluencies, by the third grade the contribution decreases to about half. This strengthens the assumption that with the development of the different fluency skills, they rely more on specific abilities in each domain, but in the first stages of acquiring these skills, the EFs play a more significant part. In addition, while EFs in kindergarten contribute the domain-specific fluencies in reading and arithmetic in the first and third grades, their contribution to the shared variance of these two fluencies was larger; these results suggest that EFs might contribute, in addition to the domain-specific fluency, to a general factor of fluency.

A substantial body of scientific evidence demonstrates that fluency in both reading and arithmetic builds upon similar information processing [90,98] and cognitive strategies and procedures [20]. As previously stated, reading and simple arithmetic calculations entail the processing of symbolic representations (e.g., letters, numerals, and mathematical symbols) [10,12]. Learning to read and perform arithmetic calculations rely on similar cognitive strategies, such as chunking, which is required for the efficient recognition of common letter combinations and number equations [99,100,101]. Adequate implementation of common serial strategies [20] at the advanced stage of skills acquisition and gradually increased automatization facilitate the rapid and accurate retrieval of words and basic arithmetic facts [23,28] in fluency-timed measures. In this regard, EFs were found to support the generalization and application of cognitive mechanisms across different academic tasks [102]. Considering the above cumulative evidence with comparable scientific support [103,104,105], we have reached the conclusion that EFs facilitate the transfer of similar processes and possibly act as a linking chain that promotes the cross-domain developmental association between fluency in reading and arithmetic.

Neuroimaging studies might also provide valuable insights into the intertwined nature of the associations between fluency performance and EF skills. Accordingly, overlapping network regions, which are activated during tasks requiring the rapid retrieval of phonological representation and numerical information [106], were also found to support EF skills [107,108]. As skill improves within one domain, it likely instigates neuroplastic changes in the shared networks, which enhance cognitive abilities that extend to the other domain [109]. These neuroplastic changes underscore the interdependence of fluency skills in reading and arithmetic and underscore the pivotal role of EFs in facilitating their reciprocal benefits.

In conclusion, the current study illuminates the interconnected nature of fluency, a fundamental skill, in reading and arithmetic, underscoring the critical role of early EF skills in their development and cross-domain relationships. These findings further highlight the importance of cognitive processes in facilitating the acquisition of fluency and the transfer of skills across both academic domains, mainly in the first years at school.

### 4.4. Limitations and Implications

Despite the significant findings obtained in this study, there are several noteworthy limitations that should be taken into consideration in future research. While the current study benefited from a large sample, it is important to acknowledge that the sample was limited to typically developing children in a specific phase of fluency skill development. Moreover, the findings are limited to the specific characteristics of the Hebrew orthography, known for its relatively consistent and shallow structure. To provide a comprehensive understanding of the developmental trajectories of these skills and how the associations between fluency in reading and arithmetic may change or become more stable over time, future research should explore diverse groups (e.g., children with learning difficulties) with different cognitive profiles across a wider age range (e.g., older children and adolescents) and orthographies to better capture the heterogeneity within the population.

In addition, our study primarily focused on examining the role of EFs in the connection between the two fluency measures. We did not consider other factors that could potentially influence the cross-domain associations. To gain a more comprehensive understanding of these relationships, it would be beneficial to explore additional measures that assess different aspects of reading and math-related skills, such as comprehension or problem-solving. Moreover, incorporating alternative variables that may interact with EFs, such as motivation and attentional control, could provide further insights into the complexities of the cross-domain relations. Another unresolved inquiry pertains to the underlying cognitive processes and neural mechanisms that facilitate the transfer of skills between these domains, with potential avenues for investigation through the utilization of neuroimaging techniques or experimental manipulations.

Future studies should also examine the development of the EFs along with the fluencies. It seems that the contribution of EFs from kindergarten decreases with age; it could be due to the rapid development of these abilities. It could be that the EFs measured in the third grade would have a larger contribution to the shard variance of reading and mathematical fluencies. Furthermore, additional studies need to investigate the putative mutual effect between academic fluency and EF in order to better understand this relationship.

The results of the present study have imperative theoretical and practical implications. Theoretically, our study advances our knowledge about the delicate interplay between fluency in reading and arithmetic by highlighting the role of early EF skills. Developmentally, the data presented in this study shed light on the process of fluency acquisition by recognizing potential variations in the developmental trajectories of fluency in reading and arithmetic. Practically, educators and curriculum designers can consider incorporating cross-domain instruction that explicitly connects reading and arithmetic, leveraging the shared cognitive processes and strategies involved in both skills. Effective integration and incorporation of EF skill training (such as WM, inhibitory control, and cognitive flexibility) might foster the successful development of fluency in both academic domains among young learners. Ultimately, proactive assessment and identification of EF skills among kindergarten children with difficulties can inform targeted interventions, which eventually might narrow the achievement gap and promote more equitable educational outcomes for all learners.

## Figures and Tables

**Figure 1 behavsci-14-00835-f001:**
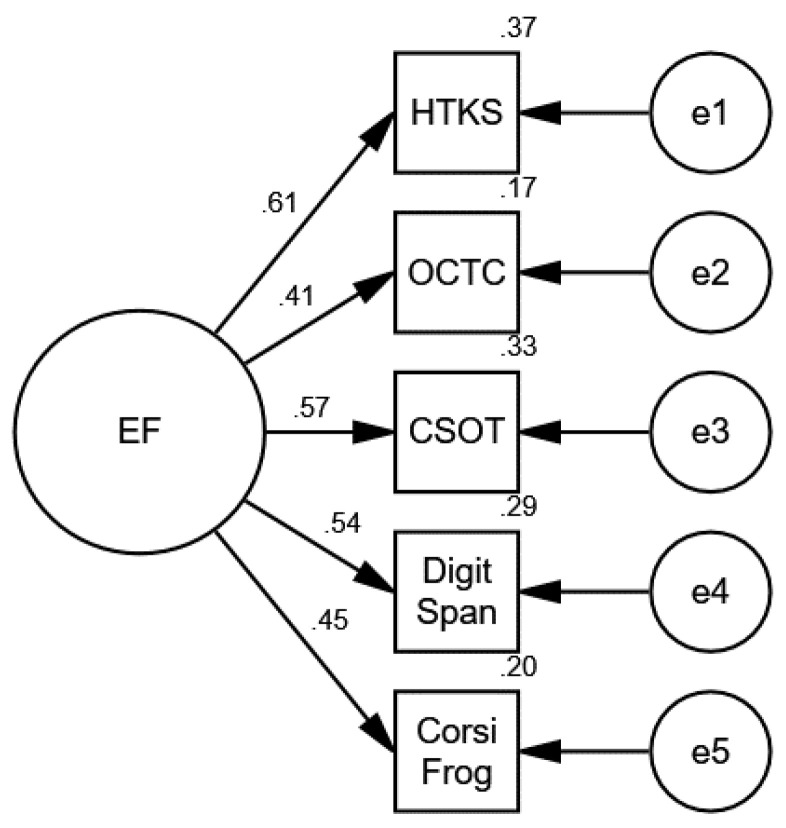
Confirmatory factor analyses (CFAs) of the observed EF variables in kindergarten.

**Figure 2 behavsci-14-00835-f002:**
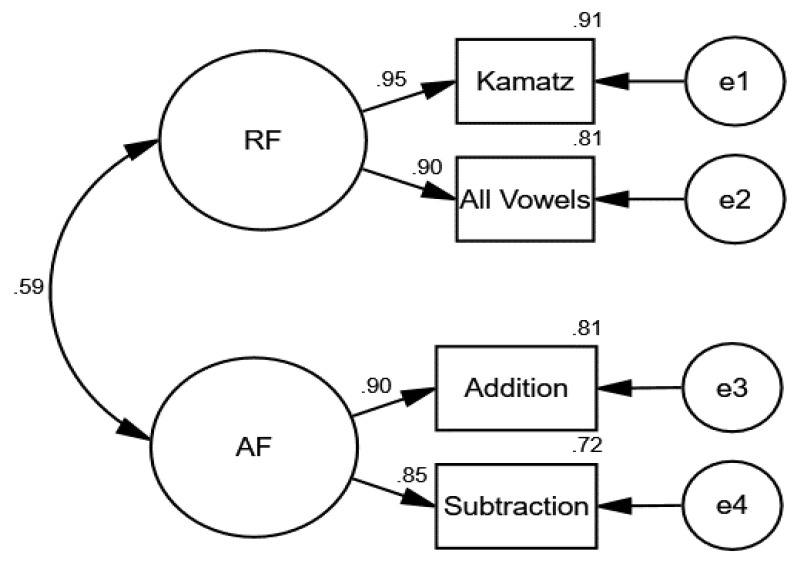
Confirmatory factor analyses (CFA) of the observed reading and arithmetic fluency variables in the first grade.

**Figure 3 behavsci-14-00835-f003:**
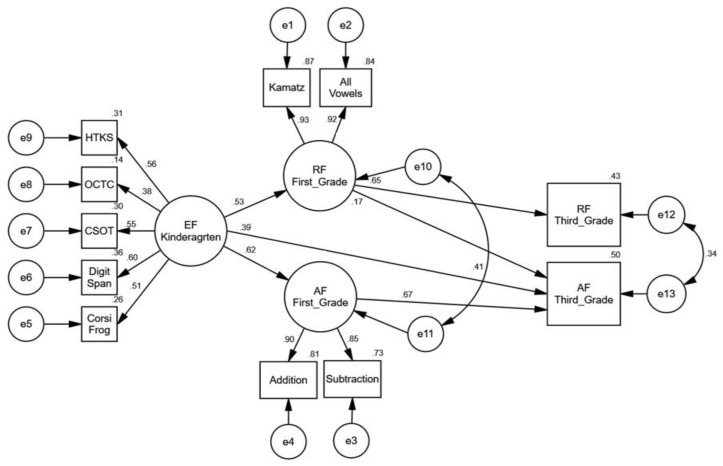
Structural equation model (SEM) of longitudinal relations between kindergarten EFs and fluency development in reading and arithmetic among first- and third-grade students. Nonsignificant correlation between initial status and growth factor is not shown.

**Figure 4 behavsci-14-00835-f004:**
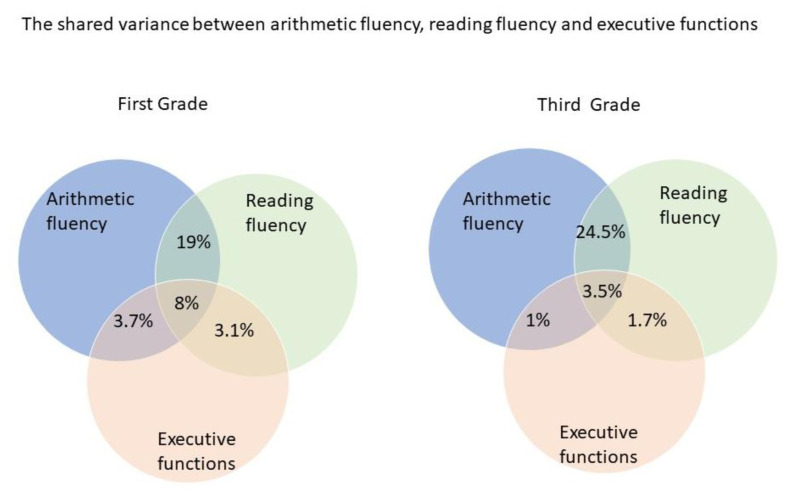
The percent of the shared variance between reading fluency, arithmetic fluency, and kindergarten executive functions in the first and third grades.

**Table 1 behavsci-14-00835-t001:** Sample Descriptive Statistics and Correlations Between EF Composite Scores and Reading and Arithmetic Fluency in the First and Third Grades.

Observed Variables	1	2	3	4	5
1. Reading fluency first grade	-				
2. Reading fluency third grade	0.619 **	-			
3. Math fluency first grade	0.517 **	0.375 **	-		
4. Math fluency third grade	0.460 **	0.533 **	0.666 **	-	
5. EF (kindergarten) composite score	0.333 **	0.211 **	0.363 **	0.229 **	-
N	708	708	708	708	708
Minimum	0	13	0	0	−2.10
Maximum	49	78	30	122	2.62
Mean	16.05	39.79	16.58	56.02	0.02
Standard deviation	10.02	11.31	5.70	17.56	0.79
Skewness	0.982	0.315	−0.026	0.369	−0.531
Kurtosis	0.855	0.427	−0.432	0.504	0.444

** Correlation is significant at the 0.01 level (2-tailed). N = the number of participants assessed for each specific measure who were part of the SEM model.

## Data Availability

The data are part of a large longitudinal study and will be available upon request.

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
