# Peer review of "Bridging the Connection between Fluency in Reading and Arithmetic"

_behavsci, 2024, doi:10.3390/bs14090835_

Round 1

Reviewer 1 Report

Comments and Suggestions for Authors

This manuscript reports on a longitudinal study with a large sample, tested for executive functions in kindergarten and for reading fluency (in a language with transparent orthography) and arithmetic operation fluency in grades 1 and 3. The main findings indicate correlations slightly above .50 between reading and arithmetic fluency in each grade, moderate correlations between executive functions in kindergarten and later academic skills, and an asymmetric pattern of longitudinal effects. In particular, executive functions in kindergarten affected both reading and arithmetic in grade 1, but only arithmetic in grade 3; and reading in grade 1 affected arithmetic in grade 3, but not vice versa. These results are interesting and worth reporting. Subsections 4.1 and 4.2 of the final Discussion are clear and fine. However, there are three major issues that the authors need to address in a revision.

(1) The mediation analyses as currently framed are illogical. I recognize that it is technically possible to run a mediation analysis in which a variable X measured at time 1 is defined as a mediator between variable Y measured at time 2 and variable Z measured at time 2; actually, this is what the authors did. However, positing this pattern of direct and indirect effects implies assuming that variable Y at time 2 can have an effect on variable X at time 1. In other words, it implies assuming that causality could go backward in time. Obviously, this contradicts the very concept of causality. Therefore, I strongly suggest removing point (c) in the subsection The present study (lines 186-189), lines 298-305 in the subsection Data analysis plan, the whole subsection Mediation models (lines 398-433) in the results, point 3 (lines 442-443) in the first paragraph of the Discussion, and the whole subsection 4.3 (lines 506-551) in the Discussion.

Instead, it would make sense asking whether and to what extent earlier executive functions contribute to (i.e., account for) the correlation between reading and arithmetic fluency at later ages. For instance, this could be done by computing partial correlations and examining to what extent partialling out earlier executive functions reduces the size of correlations between simultaneous or time-lagged measures of reading and arithmetic. Alternatively, and perhaps more elegantly, the same issue could be explored by means of a series of two-step regression analyses that enable distinguishing, within the variance shared by reading and arithmetic, the proportion of variance that is or is not shared also by earlier executive functions.

(2) A clearer and more detailed description of the participants is required. This would be definitely useful also if the research had been carried out in another country, but it is even more necessary given that it was done in a peculiar context as Israel, and the sample is qualified at various points in the manuscript as nationally representative. It is not specified why the authors consider it nationally representative. However, as we all know, the social situation in Israel is quite peculiar. There are apartheid rules that discriminate against people of Arabic ethnicity or Palestinian origin; there are geographic areas characterized by endemic conflict, and illegally occupied areas beyond the borders threatened by violent, abusive colonizers; moreover, the State grants one particular religion a privileged status, also including specific educational conditions for the haredim caste. Given that this research was carried out at Haifa, probably the sample did not include children in illegally colonized areas, but if the sample is nationally representative, I suppose that it includes some proportion of children from (probably discriminated) ethnic or religious minorities, and perhaps also from areas with more intense conflict. Therefore, a clear and rather detailed description of the sample composition is required. Moreover, in case the proportion of children from ethnic or religious minorities is substantial, it might be worthwhile (and I leave to the authors to judge whether it is more informative doing so) to include in the data analyses some control variable or even independent variable that reflects the social composition of the sample.

(3) The report of factor analyses is a bit confusing. On lines 288-289 the authors announce a “confirmatory factor analysis (with varimax rotation)”, but varimax rotation is not used in CFA. It is only used in exploratory factor analysis and principal component analysis. Also on line 307 the title of subsection 3.1 reads “Confirmatory factor analyses of the observed executive functions…” but then a principal component analysis is reported (however, without reporting the eigenvalues) on lines 310-314 and Table 1. A confirmatory factor analysis of the same variables is reported on lines 317-319 and Figure 1 (and then, another CFA of reading and arithmetic measures on lines 322-326 and Figure 2). The confusion is calling “confirmatory” also an exploratory analysis. Furthermore, it is often unnecessary to run both an exploratory and a confirmatory analysis on the same variables. If one really needs to do so (e.g., for examining the structure of a new test with several components) then one can run an exploratory analysis on half of the sample and test it for confirmation in the other half. However, that procedure does not seem necessary here. Actually, I think that the exploratory principal component analysis is not really needed. There are only five variables, of which only one is a measure of inhibition – the first executive function that, according to the literature, differentiates from working memory as a separate factor at an age of about 4½ or 5 years. With these few variables, and only one that is an inhibition measure, it is a priori unlikely that more than one factor could be extracted. Therefore, I would suggest reporting only confirmatory factor analyses (omitting the PCA and taking care to avoid confusion in terminology).

Other, minor points:

Line 59, specific-language should be language-specific.

Line 61, affect should be effect.

Lines 75-77, sentence “In addition … separately”, your research does not do this. This sentence can be dropped.

Lines 79-81, the sentence “Second … domains” can be replaced according to the changes suggested above at point (1).

Lines 88-92, the three functions studied by Miyake et al (2000) are inhibition, shifting, and updating. If you consider instead the triad inhibition, working memory, and flexibility, this was proposed for instance by Diamond (2013).

Lines 213 and 232, I presume that these test-retest reliabilities are cited from some previous research; please provide the reference.

Lines 221 and 228, please specify how many sequences per block.

Lines 224 and 263, were these Cronbach’s alphas computed on your data or reported from some other research?

Line 232, “(capacity)” – this brief mention of capacity is unclear.

Regarding the WM tests, I have no objection to calling the first visual-spatial and the second auditory, because these are the codes that they probably involve, but the third cannot be called simply “auditory”, because ordering words by the size of the objects they represent involves using semantic codes.

Lines 236-237, if the blocks range from two to seven items, then they are not seven blocks, but six.

Line 239, what reliability index, test-retest, odd-even, or Cronbach?

Line 246, plan, perhaps it was meant plane.

Lines 262-263, this description of the scoring is not clear.

Table 2, in the descriptive statistics please report also skew and kurtosis. These are useful for understanding whether there was any serious discrepancy from normal distributions.

Figure 3, some of the numbers are placed in ways that one cannot understand clearly what they refer to. In particular, is the coefficient from executive functions to grade 3 arithmetic fluency .39 or -.15? Is the coefficient from grade 1 reading fluency to All Vowels .92 or .28? And what is the meaning of the numbers placed near to a corner of each square?

Line 391, I agree that the fit of the model is good, but “excellent” might perhaps be excessive. What was the confidence interval of the RMSEA?

Lines 395-396, why “additionally”?

Line 504, rather than “the development”, I would say that “the automatization process” is domain specific and not general. A subtle conceptual difference, but not irrelevant, given that domain general executive functions contribute to the development of both academic skills.

Comments on the Quality of English Language

The English language is ok.

Reviewer 2 Report

Comments and Suggestions for Authors

Thank you for the opportunity to review this well-written manuscript.

The manuscript focuses on the development of fluency in reading and arithmetic during the first years of schooling. It explores the association between both forms of fluency by considering the role of a common underlying cognitive mechanism, namely executive functioning (EF).

The relationship between reading fluency and arithmetic fluency, as well as the relationship between EF and both academic skills, is theorized to be bidirectional. Specifically, the authors argue that while stronger EF skills tend to support the development of academic skills, academic activities also provide opportunities for practicing EF. According to the authors, this mutual influence positions EF as a potential mediator in the association between reading and arithmetic fluency.

The theoretical question is relevant, and the longitudinal nature of the study, as well as its sample size, makes this an opportunity to obtain interesting answers. However, I believe there are several drawbacks that make the manuscript less straightforward than desirable and require major revision.

Considering the SEM (Figure 3), I believe two aspects of the model need further discussion or reflection:

a)       The negative impact of EF on Arithmetic Fluency (3rd grade): The authors mention this result on Lines 487-490 of the Discussion but do not address the negative nature of the correlation. Please revise this section, as the negative impact of EF requires a completely different interpretation.

b)      The correlation between error terms associated with fluency in both years. For the fluency measures taken in 1st year, this indicates that fluency skills are correlated with other aspects beyond EF. For the fluency measures taken in 3rd year, this indicates that fluency skills are correlated with other aspects beyond the fluency levels measured in the 1st year and the previous EF. The moderate magnitude of these correlations suggests the presence of common factors.

Regarding the mediation analysis, I believe it is not well-conceived.

a)       A variable measured in kindergarten cannot be used as mediator of the relationship between two measures taken in first grade. The authors refer to Baak’s study (lines 145-149; 517-519), where EF measured in the 1st grade was used to mediate the relationship between Math in KG and Reading in the 5th grade. So, Baak maintains a logical temporal sequence, while the mediation models tested in this manuscript violate the temporal sequence of the variables. If the authors argue that EF supports the development of fluency in both academic domains (simultaneous, concurrent measured), I believe this might not be a mediation process; a common factor model is likely more appropriated than the idea of mediation. This common factor model (EF as an antecedent of both fluency measures) is statistically equivalent to the ones tested in section 3.4. However, please, note that this type of model is already tested in the SEM presented in Figure 3. Furthermore, when the authors explain the mediation effect (Lines 519-522), even they do not adhere to a mediation argument but rather use a “common factor” argumentative line.

b)      Second, I believe it is not appropriate to test bidirectional effects by reversing the direction of the direct effect in a mediational model. As Thoemmes clearly states (2015; https://doi.org/10.1080/01973533.2015.1049351): “Reversing arrows in the classic tri-variate X-M-Y mediation models as a test to check whether one mediation model is superior to another is inadmissible”. The correct approach for testing reciprocal relations with SEM involves the use of instrumental variables (see: Wong & Law, 1999, doi:10.1177/109442819921005).

Finally, in the Introduction the authors refer to the topic of a putative mutual effect between academic fluency and EF. However, such topic is not explicitly addressed in Discussion (I believe the study design does not allow to explore this topic). Considering the theoretical relevance of the topic, perhaps the authors could present it as a limitation of the study.

These are my main concerns, that – if acknowledged by the authors – will require a deeper revision of the manuscript. Please, find below a list of my minor concerns.

Minor:

Table 2. The sample sizes displayed here are not clear. What is the meaning of N in the table? These values cannot be interpreted as the sample size on which correlations were computed. Consequently, I interpret them as the number of participants assessed for each specific measure. For instance, variable 3 (“Math fluency first grade”) was assessed in N = 804 children. However, in lines 194-195 the authors have stated that only 798 children participated in Time 2 (first grade). Please, clarify. Perhaps a note below the table should specify the sample sizes range used for the correlation in each time point.

Line 61: Please, replace "bidirectional affect" with "bidirectional effect"

Line 266: Please, explain more clearly what is the “kamatz-patach” subtest. The provided information (“a single-vowel test”) is not clear: Are the children naming single vowels? If yes, this is not reading. Are the children reading single-vowel words?

Lines 288: The authors refer to “confirmatory factor analysis” but I believe they intend to say “exploratory factor analysis”.

Lines 288-290; 310-315: I believe that the exploratory factor analysis is redundant, not necessary, and even misleading. It is highly recommended not to run a confirmatory analysis on the same data used for the exploratory analysis that informed the measurement model under confirmation. I do not know if this is the case here (that the exploratory results informed the confirmatory measurement model) – what will be wrong. But I believe that the confirmatory model was theoretically expected (as supported by the Introduction) and does not need an exploratory preliminary factor analysis.

Lines 359: The authors stated they began by fitting a saturated model. However, the test of the model involved 36 df, so the model is not saturated. Perhaps they intended to say that the structural part of the model (the relationship between the latent variables and the fluency outcomes at the 5th grade) is saturated. Please, revise accordingly.

Line 360: The authors stated the measurement model demonstrates goof fit. Are they referring to the whole model or just the measurement part? Please, clarify.

Line 390: I believe the information in parenthesis refers to the comparison test between models. If yes, please consider using the delta symbol before X2 and p-value to indicate, as usual, that this is a comparison between nested models.

Round 2

Reviewer 1 Report

Comments and Suggestions for Authors

The revised version of this manuscript is much improved with respect to the original. A few minor points could still be revised.

Lines 86-90. As already noted in the original submission (it was on lines 88-92), the three functions studied by Miyake et al (2000) are inhibition, shifting, and updating. If you consider instead the triad inhibition, working memory, and flexibility, this was proposed by Diamond (2013). It is not just a matter of different terminology for the same constructs. Those models of executive function are partly different. Updating is correlated with, but distinguishable from working memory capacity (see Himi et al, 2021, Journal of Intelligence; Panesi et al, 2022, Journal of Intelligence; Redick & Lindsey, 2013, Psychonomic Bulletin & Review). Flexibility is a broader concept than shifting (see Deak & Wiseheart, 2015, Journal of Experimental Child Psychology; Ionescu et al, in press, Journal of Cognition; Morra et al, 2018, Journal of Experimental Child Psychology). Here the authors could simply acknowledge that there are some differences between Miyake’s and Diamond’s models, and that they are adhering to Diamond’s model.

Lines 216 and 233 (in the original, 213 and 232), if the test-retest reliabilities are from this study, as the authors state in their response to reviewers, then I infer that some (or all?) participants were tested twice on the HTKS and the backward digit span. This should be briefly mentioned in the Method.

The details of Figure 3 that were unclear in the original submission are still unclear. On the arrow from EF to third-grade AF, I can see two numbers (-.15 and .39), which of them is the value of this “arrow” (regression parameter) and what is the meaning of the other number? Same for the two numbers (.28 and .92) on the arrow from first grade RF and All vowels. As for the numbers at the corners of the squares, I appreciate that they are squared multiple regressions as the authors explain in their response to reviewers, but the readers of the journal cannot appreciate this if the authors do not mention it in the figure caption.

Line 383, here “an excellent fit” seems too emphatic. The indexes of goodness of fit are not strikingly different from those reported on lines 352-353 for a less parsimonious model, which were qualified as “good fit indices”. I would say “good fit” also on line 383, instead of “excellent”. (In my previous review I mentioned the confidence interval of RMSEA because if the C.I. included 0 the word “excellent” could be justified. Here the C.I. is not reported; however, based on the RMSEA value and the N size, I would bet that it does not include 0. Anyway, this detail is not necessary here. A less emphatic adjective is sufficient as a revision.)

Lines 414-415 and 431-432, “with executive function”, here I would add “with executive function measured one year earlier” and "with executive function measured three years earlier”, respectively.

Line 414, the expression 3-4% is too vague, it could mean 3.1% or 3.9% or anything in between. A more precise value with one decimal point would be clearer.

Line 431, same for the expression 1-2%

The contribution of EF at kindergarten to the correlation between AF and RF at first/third grade can be assessed as the ratio of variance shared with EF to the variance shared between AF and RF. For example, let’s suppose that at first grade the variance shared also with executive function was 3.5% and the total variance shared between AF and RF was (27.0% + 3.5%); in this case, the contribution of early EF to the correlation between AF and RF would be 3.5/30.5 = 11.5%. (Of course, this is a fictitious example; the computation should be made with the actual values.) This detail could be reported in the text, and perhaps also briefly considered in paragraph 4.2 or 4.3 of the final discussion.

Lines 487-488, why “negatively”? The correlations in table 2 clearly show a positive correlation of .227 between early EF and third-grade arithmetic fluency. Consequently, the whole passage on lines 489-497 is unclear to me. (And, as mentioned above, there are two numbers, .39 and -.15, on the arrow between EF and third-grade AF, and it is unclear which of them is correct; based on the correlations I would bet that .39 is the correct one, but of course I cannot be sure.)

Please also check the manuscript for occasional typos.

I think that, when these few details are corrected, the manuscript will be ready for publication.

Author Response

all comments are in the attached file

Reviewer 2 Report

Comments and Suggestions for Authors

Dear Authors,

Thank you for your efforts in addressing my previous suggestions. I appreciate the work you've put into revising the manuscript.

However, I have a few additional points of concern:

1. Spelling and Typographical Errors: I strongly recommend running a spelling and grammar check, as there are numerous typographical errors throughout the manuscript (e.g., “regarrisons” instead of “regressions” and “The reuslt emphsize that eraly Ef’s contribute,” among others). These errors can detract from the clarity and professionalism of the work.

2. Clarity of Details: There are several details throughout the manuscript that require clarification. These ambiguities may confuse or distract an attentive reader, so please see below some aspects that I think need to be clarified.

3. Main Concerns: While the substitution of the mediation analysis with a hierarchical regression analysis is noted, it still does not fully align with the goals outlined for the study. The limitations of the study's design and the chosen analytic procedure hinder the ability to draw the intended conclusions. Please, check below for details.

Section 1.3 “Executive Functions as a Mediator”:

This section now feels somewhat out of place, as the role of executive functions (EF) as a mediator is not empirically explored in the revised manuscript. Although the concept that EF may influence fluency and also improve with fluency training is interesting, the study's design does not support an investigation of this bidirectional effect. I suggest revising this section to better align with the goals of the current version of the manuscript. One possibility is to integrate some of these ideas into the Discussion section. However, I believe that dedicating an entire section to the mediation role of EF is excessive under the current study design.

Bidirectionality of Reading and Arithmetic Fluency:

The manuscript refers also to the bidirectional influence between reading fluency and arithmetic fluency, such as in lines 152-155, where you state: “Thus, the main goal of our study is to elucidate and directly investigate whether and how early EF skills, before school transition, affect the relationship between fluency in reading and arithmetic from a bidirectional perspective.” I believe the study's design does not support to extract conclusions about such bidirectional influences. The regression analyses conducted only provide estimates for specific and shared variance and do not clarifications regarding EF impact on the reciprocal relationship between reading and arithmetic fluency. These sections should be revised to accurately reflect the study's limitations. In my view, the current design does not allow for the exploration of bidirectional influences among the variables.

Reformulated Third Research Question:

The third research question - To what extent do earlier executive functions contribute to (i.e., account for) the connection between reading and arithmetic fluency at later ages?” - appears redundant after the second research question. As it stands, this question seems implicitly nested within the second research question. It may be more appropriate to either delete this third question or reconsider its placement.

Confirmatory Factor Analyses (CFAs):

The two CFAs, particularly the second one, do not add substantial information, as they are part of the measurement model in the SEM analysis. However, if you decide to retain them, I recommend reporting a model-based reliability index, such as the omega coefficient, for the EF model.

EF Score:

The EF score is reported as a z-score, yet the mean is 0.9 and the SD is 0.6, which is unexpected. If this discrepancy is due to the participants being a subsample (N = 764) of the original kindergarten sample (N = 1,142), where the EF score had the expected mean of 0 and SD of 1, please clarify. If this is the case, it could indicate a biased subsample with a higher level of EF than the drop-out participants. However, if the EF scores were available for the complete sample, why are the results in Table 1 based on N = 764 and not on the full sample of N = 1,142? I would understand if Table 1 reported only the subsample used for the SEM model (N = 708?), but that does not appear to be the case. Therefore, clarification is needed regarding the subsamples actually used in the analyses.

Hierarchical Regression Models:

The relevance of the information in this section is not totally clear. Furthermore, if the goal was “to examine to what extent early kindergarten EF contributes to the shared variance between reading and arithmetic fluency” (line 393), the chosen analytical procedure only estimates the specific contribution of EF, excluding the shared variance between RF and AF. This approach underestimates EF's total contribution, as EF likely influences both fluencies, with its impact included in the shared variance. If you retain this analysis, consider reporting the total contribution of EF, differentiating its impact on both the shared and specific variances of each fluency measure. Additionally, you might estimate the shared variance between RF and AF not accounted for by EF. This information could be effectively presented in a table or a Venn diagram.

Discussion Section:

Naturally, some parts of the Discussion should be updated to reflect the changes in results and analysis.

Once again, I appreciate your hard work on this manuscript and look forward to your revisions. Please see some minor details below:

Line 282:  Consider rephrasing to: "In both arithmetic subtests, children are instructed to complete as..." to clarify that the task was similar in both subtests.

Table 1: The information in Table 1 is still unclear. The authors mentioned that there were 1,142 participants in kindergarten, 798 in the 1st grade, and 716 in the 3rd grade. However, these numbers do not match those reported in the table. For example, EF composite scores were calculated for 764 participants (corresponding to only two-thirds of the original sample). What happened to the remaining third? Additionally, how can the descriptive statistics for the 3rd grade be based on N = 804, when the number of participants is reported as 798? Finally, since the SEM model and regressions must be based on the subsample where all measures are available (N = 708), perhaps the correlations reported should be based on this subsample rather than on samples of varying sizes.

Given these points, I also suggest the authors indicate the number of participants involved in each of the CFAs.

Line 383: Please correct the figure numbering.

Comments on the Quality of English Language

I strongly recommend running a spelling and grammar check, as there are numerous typographical errors throughout the manuscript (e.g., “regarrisons” instead of “regressions” and “The reuslt emphsize that eraly Ef’s contribute,” among others). These errors can detract from the clarity and professionalism of the work.

Author Response

all comments are in the attached file

Round 3

Reviewer 2 Report

Comments and Suggestions for Authors

I want to thank again the authors for their efforts in addressing my previous suggestions. However, I still have some concerns regarding the section describing the hierarchical regression results.

In the sentence "The results indicate that 27% of the variance in reading fluency is shared only with arithmetic fluency", I suggest deleting the word ONLY because 27% is the total variance shared between reading fluency and arithmetic fluency, but it can also be shared with a third variable (or other variables). In this case, part of this 27% is also shared with EF.

I do not understand this sentence: "In addition another 3.3 % (reading with arithmetic) and 3.7% (arithmetic with reading) of shared variance, is also shared with executive function measured one year earlier." Perhaps the authors are referring to the values R2 = 0.03 and R2 = 0.04 previously mentioned. If yes, these percentages correspond to the variance uniquely shared between EF and each one of the fluency measures and not to the shared variance between the three variables.

"This means that out of the total sheared variance between the two fluencies, EF in kindergarten explains 11% of the shared variance between reading and mathematical fluency and 12.1% of the variance of mathematical and reading fluency in the first grade." I am not sure about this interpretation. From Table 1 we know that EF and Reading Fluency correlate r = .333. So, we can say that EF shares 11.1% (r2 = 100*.333*.333 = 11.1%) of the variance of Reading Fluency. But we cannot say that "EF’s in kindergarten explains 11% of the shared variance between reading and mathematical fluency". Indeed, what we can say is that “EF’s in kindergarten explains 11% of the variance of reading fluency”. Making some raw calculations, I will say that "EF’s in kindergarten explains 8% of the shared variance between reading and mathematical fluency". Furthermore, it does not make sense to say: "EF’s in kindergarten explains 11% of the shared variance between reading and mathematical fluency and 12.1% of the variance of mathematical and reading fluency in the first grade." If EF is explaining the shared variance between reading and mathematical fluency, the percentage of explained variance shared between reading and mathematical fluency must be exactly the same of the percentage of explained variance shared between mathematical and reading fluency.

Please, consider the attached file with a Venn diagram representing the shared and unique variance between the three variables.

So, I suggest the authors to review the paragraph into something like:

"The results indicate that 27% of the variance in reading fluency is shared with arithmetic fluency. However, out of the total sheared variance between the two fluencies, EF in kindergarten explains 8%. The unique contribution of EF to either reading fluency and arithmetic fluency in the first grade is clearly smaller (namely, 3.1% and 3.7%)." Of course, the exact percentages should be calculated based on the complete data; I am using correlations from table 1.

These results suggest that EF might have a main contribute for a general factor of fluency, rather than to the specific characteristics of reading fluency and arithmetic fluency.

The same reformulation can be done in relation to the 3rd grade fluency measures:

"The results indicate that 28% of the variance in reading fluency is shared with arithmetic fluency. However, out of the total sheared variance between the two fluencies, EF in kindergarten explains 3.5%. The unique contribution of EF to either reading fluency and arithmetic fluency in the first grade is smaller (namely, 1.0% and 1.7%)."

Please, consider not to call "EF z-score" (table 1) but instead of EF composite (since the average of z scores is not a z-score itself - the mans is still 0 but the SD will not be 1, as expected for a z-score).

Please, consider revising typos (line 90: "Dimond"; line 409: "sheared").

Thank you again for your patience.

Comments on the Quality of English Language

Spell checking, please 

Author Response

letter is attached
